# Preparation of 6-Mercaptopurine Loaded Liposomal Formulation for Enhanced Cytotoxic Response in Cancer Cells

**DOI:** 10.3390/nano12224029

**Published:** 2022-11-16

**Authors:** Alam Jamal, Amer H. Asseri, Ehab M. M. Ali, Afnan H. El-Gowily, Mohamed Imran Khan, Salman Hosawi, Reem Alsolami, Tarek A. Ahmed

**Affiliations:** 1Department of Biochemistry, Faculty of Science, King Abdulaziz University, Jeddah 21589, Saudi Arabia; ajamal0015@stu.kau.edu.sa (A.J.); ahasseri@kau.edu.sa (A.H.A.); mikhan@kau.edu.sa (M.I.K.); shosawi@kau.edu.sa (S.H.); 2Centre for Artificial Intelligence in Precision Medicines, King Abdulaziz University, Jeddah 21589, Saudi Arabia; ramalsolami@kau.edu.sa; 3Division of Biochemistry, Department of Chemistry, Faculty of Science, Tanta University, Tanta 31527, Egypt; afnan.hamdy@science.tanta.edu.eg; 4Department of Medical Laboratory Sciences, Faculty of Applied Medical Sciences, King Abdulaziz University, Jeddah 21589, Saudi Arabia; 5Department of Pharmaceutics, Faculty of Pharmacy, King Abdulaziz University, Jeddah 21589, Saudi Arabia

**Keywords:** 6-mercaptopurine, liposomes, human cell lines, IC_50_, apoptosis, cell cycle phases

## Abstract

6-Mercaptopurine (6-MP) is a well-known immunosuppressive medication with proven anti-proliferative activities. 6-MP possesses incomplete and highly variable oral absorption due to its poor water solubility, which might reduce its anti-cancer properties. To overcome these negative effects, we developed neutral and positively charged drug-loaded liposomal formulations utilizing the thin-film hydration technique. The prepared liposomal formulations were characterized for their size, polydispersity index (PDI), zeta potential, and entrapment efficiency. The average size of the prepared liposomes was between 574.67 ± 37.29 and 660.47 ± 44.32 nm. Positively charged liposomes (F1 and F3) exhibited a lower PDI than the corresponding neutrally charged ones (F2 and F4). Entrapment efficiency was higher in the neutral liposomes when compared to the charged formulation. F1 showed the lowest IC_50_ against HepG2, HCT116, and MCF-7 cancer cells. HepG2 cells treated with F1 showed the highest level of inhibition of cell proliferation with no evidence of apoptosis. Cell cycle analysis showed an increase in the G1/G0 and S phases, along with a decrease in the G2/M phases in the cell lines treated with drug loaded positively charged liposomes when compared to free positive liposomes, indicating arrest of cells in the S phase due to the stoppage of priming and DNA synthesis outside the mitotic phase. As a result, liposomes could be considered as an effective drug delivery system for treatment of a variety of cancers; they provide a chance that a nanoformulation of 6-MP will boost the cytotoxicity of the drug in a small pharmacological dose which provides a dosage advantage.

## 1. Introduction

Cancer kills 16% of the world’s population, making it the second leading cause of mortality after heart disease. Cancer treatment with radiation and chemotherapies cause side effects such as anemia and hair loss, as well as immune system imbalance. Some patients do not react to treatment because of changed gene patterns that results in chemotherapy resistance, which causes substantial side effects [1].

6-Mercaptopurine (6-MP) is a medication that suppresses the immune system and decreases cell growth by competing with adenine for DNA synthesis and transcription, hence reducing cell growth. It is created by the formation of 6-thioguanine nucleotides, which causes cancer cells to die and disrupts DNA replication and RNA transcription [2]. Furthermore, 6-MP inhibits ATP generation and induces programmed cell death, limiting cell proliferation and inhibiting cancer progression in the body [3]. 6-MP has an aqueous solubility of 0.22 mg/mL, its absorption ranges from 16–50%. Patients’ reactions to 6-MP varies as well. There are also major negative effects from using 6-MP for 2 to 3 years which leads to hepatotoxicity and myelosuppression [4]. The elimination half-life of 6-MP is ranged between 1–2 h [5] and so, drug encapsulation in an efficient carrier may be helpful to increase the drug half-life while decreasing the dose to provide more efficiency.

Lipid-based nanoparticles (NPs) have gained great attention in drug delivery. These NPs have been studied for their transport of hydrophobic and hydrophilic molecules. They have showed very low or no toxicity and an extended time of drug release due to the increase in half-life [6]. Lipid-based NPs may be classified into phospholipid and non-phospholipid vesicles. The former includes liposomes, transfersomes, ethosomes, and trans-ethosomes [7,8,9] while, the latter comprises solid lipid NPs [10], nanostructured lipid carriers [11], polymeric micelles [12,13], niosomes [14], nano-emulsions [15], and dendrimers [16]. The liposome has a structure comparable to the cell membrane, which has two phospholipid layers and is spherical in form. It comes in a variety of forms that vary depending on the quality of the phospholipids, including neutral, positive, and negative charges. Additionally, liposomes can be used to encapsulate a variety of drugs. For example, strongly lipophilic drugs are almost entirely entrapped within the lipid bilayer, highly hydrophilic drugs are entirely distributed within the aqueous compartment, and intermediate *log P* drugs are easily partitioned between the lipid and aqueous phases, both in the bilayer and in the aqueous core [17]. The Food and Drug Administration has approved the use of drug-loaded liposomes. The liposome is considered a safe chemotherapeutic carrier since it is delivered slowly, resulting in a greater effect with less toxicity [18]. Moreover, Carboxymethyl chitosan conjugates were used to encapsulate 6-MP after self-assemble in pH 7.4 phosphate buffer saline [19]. The prepared particles have been reported to modify the drug release. Furthermore, nanomaterials of chitosan encapsulated 6-MP compound recoded with iron oxide to increase the body’s retention of 6 MP and its effectiveness, and targeting the target part of the treatment reduces side effects and shortens the duration of treatment [20].

In this work, we employed liposomes as a delivery cargo for 6-MP and developed both neutral and positively charged 6-MP loaded liposomes using the thin-film hydration approach. The prepared 6-MP loaded liposomal formulations were characterized for size and drug entrapment efficiency. The anticancer activities were examined on three different human cell lines: HepG2, HCT116, and MCF-7. HepG2 was chosen as the cancer cell line to study apoptosis and cell cycle since the results of IC_50_ tests against three different cancer cells showed that IC_50_ of drugs against this cancer cell line was the lowest. Apoptosis and the cell cycle were also investigated in HepG2 cells treated with 6-MP loaded positively charged liposomes.

## 2. Materials and Methods

### 2.1. Materials

L-α phosphatidylcholine (95%) (soy), with an average molecular weight of 775.037 was obtained from Avanti^®^ polar lipids, Inc. (Alabaster, AL, USA). 6-Mercaptopurine (6-MP), tween 80, ethanol, stearyl amine, sodium hydroxide, and sodium dihydrogen phosphate were purchased by Sigma-Aldrich Company Ltd., (St. Louis, MO, USA). Dulbecco’s Modified Eagle’s Medium (high glucose) (DMEM), fetal bovine serum (FBS), HepG2, HCT116, and MCF-7 cell cultures and Penicillin-streptomycin 5000 U/mL antibiotics were procured from Thermo Fisher Scientific (Waltham, MA, USA). Trypan blue, thiazolyl blue tetrazolium bromide (MTT), and propidium iodide were provided by Merck & Co Inc. (West Point, PA, USA). The Annexin V-FITC Apoptosis Kit was obtained from Creative Biolabs (Shirley, NY, USA).

### 2.2. Preparation of 6-MP Liposomal Formulations

Thin-film hydration technique was used to develop a positively charged and neutral liposomal formulations loaded with 6-mercaptopurine [7,9,21,22]. This technique was selected as it is the most common and simple for development of all kinds of lipid-based nanoparticles, but the process must be optimized to achieve high encapsulation and homogenous size distribution. According to the formulation ingredients represented in Table 1, four different liposomal formulations were coded and prepared. A positively charged drug-loaded liposomes (F1), uncharged drug-loaded liposomes (F2), non-medicated positively charged liposomes (F3), and non-medicated uncharged liposomes (F4) were developed. Selection of the drug to lipid ratio, percentage of tween 80 and the percentage of charge inducing agent was based on our previously published work for optimization of the formulation components used to develop flexible lipid based liposomal [7,23]. A drug to phospholipid molar ratio of 1: 1.7 was used. Tween 80 and stearyl amine [CH_3_(CH_2_)_17_NH_2_] constituted 28.69% and 23.36% of the total lipid, respectively. The concentration of 6-MP was adjusted to 0.5% *w/v* based on the total liposomal formulation. A homogenous dispersion of 6-MP, L-phosphatidylcholine, tween 80, and stearyl amine was prepared in 50 mL ethanol using a CF3 2EY water bath sonicator device (Ultra-wave Ltd., Cardiff, UK). Buchi Rotavapor R-200 of Buchi Labortechink AG (Flawil, Switzerland) was used at 50 °C under reduced pressure to gradually evaporate the organic solvent (ethanol). The process was continued until a transparent fatty layer was developed on the wall of the flask. To ensure that the ethanol was completely evaporated, the flask was left overnight in a vacuum oven of Thermo Fisher Scientific (Oakwood Village, OH, USA) at 40 °C. The dried fatty layer was further hydrated, using phosphate buffer of pH 7.4, at 40 °C. Finally, the size of the produced liposomes was decreased by subjecting the formulation to sonication in a water bath for 10 min.

### 2.3. Characterization of the Prepared 6-MP Liposomal Formulations

The prepared four liposomal formulations were characterized for mean particle size, polydispersity index (PDI), and zeta potential using Malvern Zetasizer Nano ZSP of Malvern Panalytical Ltd., (Malvern, UK). All samples were examined three times. Before measuring the size and PDI, the prepared liposomal formulations were diluted with distilled water in 1:3 ratio to achieve low number density of the liposomal vesicles in the sample compartment, the effect that prevents interaction between the liposomal vesicles as previously reported [24]. Measurement of the zeta potential was conducted automatically at a scattering angle of 13°. Samples were allowed to equilibrate at 25 °C. The equilibration time was 300 s. The number of runs and scan for each sample, voltage selection and attenuation selection were all automatically set. The average of three measurements was taken.

The amount of 6-MP entrapped in the liposomal formulations was calculated indirectly as previously mentioned [7]. The prepared 6-MP loaded liposomes were centrifuged using 3 K30 Sigma Laboratory centrifuge (Ostrode, Germany) for 1 h at 20,000 rpm at 4 °C. The supernatant, containing the un-entrapped drug, was taken, filtered using an Acrodisc^®^ 0.2 syringe filter and the drug concentration in the filtered sample was determined spectrophotometrically at 332 nm. The percentage of 6-MP efficiently entrapped in the liposomes was calculated by the following equation

$$EE=\frac{Total\;amount\;of\;drug\;used-Calculated\;amount\;of\;free\;drug\;in\;the\;supernatant}{Total\;amount\;of\;drug\;used}\times 100$$


The percent of drug loading was calculated, as previously published [25], using the following equation

(1)
$$\%\;drug\;loading=\frac{\mathrm{Total}\;\mathrm{amount}\;\mathrm{of}\;\mathrm{drug}\;\mathrm{entrapped}\;}{Amount\;of\;liposomes\;}\times 100\;$$


### 2.4. MTT Assay

The Tissue Culture Unit, Department of Biochemistry, Faculty of Science, King Abdulaziz University provided the HepG2, HCT116, and MCF-7. Selected human cell lines were grown in a CO_2_ incubator at 37 °C with 10% fetal bovine serum (FBS) in DEMEM medium. An amount of 5 mL of 0.25% trypsin is added after 70–90% confluence to help detach cells. Trypan blue was used to count the cells, and the cell concentration was adjusted to 10^5^/mL. Each well of a 96-well plate was filled with 100 µL, and the plate was incubated for 24 h. Each well, media was changed to contain media with varied concentrations of 6-MP, coated positive and neutral liposomes, or uncoated liposomes. The concentrations of 6-MP free, with liposomes, or without liposomes were 50, 25, 12.5, 6.25, and 3.125 µg/mL, respectively. Each concentration was repeated four times. A total of 100 µL of the 0.5 mg/mL MTT was replaced with media containing the drug after 48 h of incubation at 37 °C. The 96-well plate was incubated for 4 h at 37 °C in the dark. The MTT was removed and replaced with 100 µL of DMSO, and the plate was left to sit for 15 min. The absorbance was measured at 595 nm using ELISA reader (Bio-RAD microplate reader, Japan) [26,27].

### 2.5. Apoptosis Assay

After a 24 h of HepG2 cell transplantation in a T 75 flask, as described in the previous section, and after adding trypsin and medium to halt its activity, cells were moved to a Falcon tube and counted as mentioned before. In a 6 well plate, 2 mL of media with 2 × 10^5^ cells was added to each well. The plate was incubated for 24 h to allow cell growth and attachment. Then, the medium was replaced with a different medium containing the same equivalent of 6-MP, 6-MP coated with a positive charge, or a positive charge liposome without 6-MP, all of which had an equivalent IC_50_ of drug. After a day, the cells were separated with 0.5 mL of 0.25% trypsin to each well of 6 well plate. The plate was incubated for 5 min at 37 °C. Each well of 6 well plate was collected in Falcon tube and centrifuged. The pellet was washed twice with PBS. An amount of 100 µL of suspended cells and 25 µL of Annexin V-FITC/propidium iodide (PI) were added in each pellet and mixed well. The tubes were incubated in the dark place for 5 min at room temperature. A total of 400 mL of binding buffer was added. The cells were detected by flowocytometer (Applied Bio-system, Waltham, MA, USA) adjacent by software to detect apoptotic and necrotic cells [28].

### 2.6. Cell Cycle Assay

HepG2 cells were cultivated 24 h in the same way as described in Section 2.1, except that, 1 × 10^6^ cells per well were added in a 6-well plate. A medium having a concentration equal to the IC_50_ of 6-MP, 6-MP coated positive liposomes, and an equivalent volume of free positive charged liposomes were then replaced after 24 h (attached cells are formed in each well). As previously noted, cells were taken from each well after 24 h. PBS was used to wash the cells twice. After the cells were suspended in 300 µL of PBS, 0.7 mL of 100% ethanol was gradually added.

The tubes were held at −4 °C for at least one hour. After centrifuging the cells, 250 µL of 50 mg/mL PI solution and 100 µL of PBS were added to the pelleted cells. The tubes were left in the dark for one hour. PI can bind to DNA and aggregate cells during each stage of the cell cycle. The target cells were run through a flow cytometer to identify the percentage of cells in each phase in cell cycle (Applied Bio-system, USA) [29].

### 2.7. Statistical Analysis

To allow for formula characterization and investigations on the viability of treated cells, results are shown as mean ± SD. Drug IC_50_ was calculated using GraphPad Prism Software (version 9.0, San Diego, CA, USA), and all the data was statistically analyzed. Applied Biosystem’s flow cytometry software automatically assessed the proportion of cells in each phase, as well as the number of apoptotic and necrotic cells.

## 3. Results and Discussion

### 3.1. Characterization of the Prepared 6-MP Liposomal Formulations

Due to their biocompatibility, non-immunogenicity, improved drug solubility, and capacity to encapsulate hydrophilic and hydrophobic drugs, liposomes have been employed in the treatment of many pathogenic conditions and to overcome side effects of drugs, especially in cancer therapy [30,31]. Drug loaded liposomes have the ability to modify the release of their payload and target specific cells or tissue.

In this work, 6-MP loaded liposomal formulations have been developed using the thin film hydration technique. The prepared vesicles have been characterized and the obtained results are illustrated in Table 1.

The average size of the positively charged liposomes loaded with 6-MP (F1) and the non-medicated positively charged formulation (F3) was found to be 574.67 ± 37.29 and 429.47 ± 24.79 nm, respectively. Neutral “uncharged” liposomes demonstrated an average size of 660.47 ± 44.32 and 538.80 ± 49.73 for the medicated and non-medicated formulations, respectively. It has been previously mentioned that liposomes are spherical vesicles that have sizes ranging from 30 nm to several micrometers [32]. Based on the vesicle size, liposomes may be classified into unilamellar or multilamellar vesicles [33]. The former may be further subdivided into small unilamellar liposomes (30–100 nm), large unilamellar liposomes (>100 nm), and giant unilamellar liposomes (>1000 nm) [34]. Multilamellar liposomes usually have size greater than 500 nm [33]. Accordingly, our results for the vesicle size infer formation of large to giant unilamellar or small multilamellar vesicles. The difference in size between the prepared non-medicated and drug loaded liposomes may be attributed to the presence of abundant liposomal lamellae which can extend to provide a large space for the encapsulation of lipophilic compounds [35]. Incorporation of stearyl amine in the formulation was found to reduce the vesicle size, probably due to its antistatic effect.

The PDI of the prepared positively charged liposomes encapsulated with 6-MP (F1) and drug free formulation (F3) was 0.543 and 0.467, respectively. The PDI of the stearyl amine free liposomes (F2) and (F4) was 0.667 and 0.711, respectively. Positively charged liposomes had a lower PDI than neutrally charged ones. Incorporation of stearyl amine in the liposomal formulation leads to its deposition on outer surface of the prepared vesicles [9], the effect that prevents aggregation of the vesicles due to the electrostatic repulsion. As a result, a monodisperse system is formed, with vesicles distributed in the medium in a more uniform manner. Our results for the PDI values indicated acceptable vesicle size distribution, since a PDI value greater than 0.7 is an indication of a very broad size distribution as previously reported by Danaei et al. [36].

Zeta potential (ZP), which refers to the electric charges of the liposomes in the medium, measures the resistance of the nano-vesicles to coalescence in the colloidal system. The liposome is highly stable in solution if the zeta potential is high. The obtained values of the ZP for F1 and F3 were +12.4 ± 1.31 mV and +9.61 ± 0.69 mV, respectively, indicating that the addition of stearyl amine resulted in slightly positive charges. On the other hand, F2 and F4 showed ZP values that were nearly equal to zero, which is an indication of a neutral charged vesicles.

The entrapment efficiency (EE) of 6-MP in the charged liposome (F1) was found to be 42.11 ± 2.07%, which is significantly lower than the corresponding neutral liposomal formulation (F2). The later showed an EE of 94.44 ± 0.56%. This behavior could be attributed to the electrostatic repulsion between 6-MP (C_5_H_4_N_4_S) and stearyl amine [CH_3_ (CH_2_)_17_NH_2_] molecules that are adsorbed on the liposome surface during the formation of the drug loaded liposomes.

### 3.2. Anti-Tumor Activity of the Prepared Liposomal Formulations against HepG 2, HCT116, and MCF-7 Cells

To determine the IC_50_ of free 6-MP and the prepared liposomal formulations, the percentage of vitality of HepG2, HCT116, and MCF-7 cells was determined after 48 h of treatment with various dosages of free and 6-MP loaded liposomes (Figure 1).

Free 6-MP had IC_50_ values of 9.6, 16.7, and 12.8 µg/mL in HepG2, HCT116, and MCF-7 cell lines, respectively. When compared to free positive liposome (F3), 6-MP coated with positive liposomes (F1) was reduced by 3.6; 3.4; and 4.4 in HepG2, HCT116, and MCF-7, respectively. When compared to free neutral liposome (F4), the IC_50_ of 6-MP coated with neutral 6-MP (F2) was decreased in HepG2, HCT116, and MCF-7 by 3.3, 2.5, and 3.6 folds, respectively. The IC_50_ values for F3 and F4 in HepG2, HCT116, and MCF-7 were (16.7, 33.9), (16.1, 37.1), and (21.5, 41.9), respectively. In all the studied cell lines, F1 showed the lowest IC_50_ value (Table 2).

The positive charges on the liposomes’ surface are expected to improve the drug absorption. When liposomes are used to deliver 6-MP to the cancer cells, its plasma half-life is increased while its toxicity is decreased. Cancer cells are more efficiently combatted by cationic liposomes, which also prevent cancer cells from growing and migrating to other organs and overcome multidrug resistance. When 6-mercaptopurine is incorporated into liposomes, continuous release at low doses and a more dispersed formulation work better against cancer cells. These compositions have the benefit of being biodegradable [37].

It has been reported that 6-mercaptopurine-9-β-Dribofuranoside (6MPR) and 6-MP encapsulated gold nanoparticles (6-MPR-AuNP) suppress the K-562 leukemia cells after 72 h of incubation at a concentration of 1.8 × 10^6^ [38]. The varied concentrations of 6-MP and 4 formulations of without or with 6-MP encapsulated with positive and neutral liposomes treated with HepG2, HCT116, and MCF-7 for 48 h equivalent to (3.125–50 µg/mL) were studied in this work (18 × 10^−6^–56 × 10^−6^ M). The IC_50_ values for 6-MP (56 × 10^−6^–98 × 10^−6^ M), F1 (27 × 10^−6^–29 × 10^−6^ M) and F2 (60 × 10^−6^–86 × 10^−6^ M) were determined in three different human cell lines. The lowest IC_50_ values for 6-MP, F1 and F2 against HepG2 were found in our data. When compared to 6-MP and 6-MP encapsulated neutral liposomes (F2), the IC_50_ of 6-MP encapsulated positive liposomes (F1) against HepG2 was low 2.1 and 2.3 folds, respectively.

In a prior work, 6-MP encapsulated poly (lactide-co-glycolide) (PLGA) (0.76 µM) had an IC_50_ against Jurkat cells that was higher than that of 6-MP (0.36 µM). 6-MP loaded PLGA toxicity to cells due to the cumulative release of 6-MP from PLGA exhibiting 90% at 48 h, may be less than 6-MP. The high value of IC_50_ against Jurkat cells is impacted by the 6-MP release rate from PLGA, which is sluggish [39].

### 3.3. Apoptosis and Necrosis of HepG2 Treated with Drug Loaded Positive Liposome (F1)

Comparing HepG2 treated with 30 µg/mL of 6-MP to untreated HepG2, necrosis rose to 1.5 while cell population decreased to 2.8. When compared to cells treated with free positive charge liposome (F3), HepG2 cells treated with 5 µg/mL 6-MP loaded positive liposome (F1) showed no symptoms of necrosis but had a lower cell population of 4.7 (Figure 2).

6-MP free releases from 6-MP-encapsulated cationic liposomes and passively enters cancer cells. Alternatively, it may enter endocytosed cancer cells, allowing 6-MP concentrations to remain greater inside the cells while also exposing 6-MP cells for a longer time, departing slowly, causing DNA to be damaged, and accelerating apoptosis [39].

In the current study, the percentage of necrosis in HepG2 treated with 6-MP was 20.3%, while it was 2.2% in F1. There was no detection of programmed cell death in either 6-MP or 6-MP loaded positively liposomes (F1). It was reported that the percentage of apoptosis and necrosis in K-562 leukemia cells treated with 6-MP and 6-MP-AuNPs for 72 h does not surpass 12 and 2.4%, necrosis and apoptosis which is a low percentage [38]. 6-MP’s mechanism of action against HepG2 is thought to be similar to that of leukemia since it inhibits cells rather than drastically increasing cell death via apoptosis or necrosis.

First quarter (lower left) = % of viable cells; second quarter (upper left) = % of necrosis; third quarter (lower right) = % of early apoptosis; fourth quarter (upper right) = % of late apoptosis. P = population of cells; and R = ratio of cells to control.

In a different study, untreated Jurkat cells exhibited 6% apoptosis while Jurkat cells treated with 0.5 µM 6-MP, and 6-MP loaded PLGA for 48 h, had a percentage of apoptosis of 8% and 15%, respectively. Increasing the concentration of 6-MP and 6-MP loaded PLGA enhanced the proportion of Jurkat cells that increase apoptosis for 48 h. 6-MP loaded PLGA maintains a high 6-MP content inside Jurkat cells for the duration of the prolonged exposure time affect the DNA damage that causes apoptosis [39].

### 3.4. Cell Cycle Analysis of HepG2 Treated with Free 6-MP and Liposomal Formulation (F1)

When compared to untreated HepG2 cells, which were arrested in sub-G1 (5.2%) phase, G0/G1 phase (29.3%), S phase (18.6%), and G2/M (46.3%), respectively, HepG2 cells treated with 6-MP at a dose of 30 µg/mL showed an increase in sub-G1 (6.7%), G0/G1 phase (37.6%) and in S phase (21.8%) and decreased in G2/M (32.8%) (Figure 3A,B and Figure 4).

HepG2 cells treated with 5 µg/mL of 6-MP coated with positive charge liposomes showed a significant increase in G1/G0 and S phases (37.5 and 11.9%) and a dramatic decrease in G2/M phases (16.1%) as compared to HepG2 cells treated with positive charge liposomes without 6-MP, which stopped cell cycle in G1/G0 and S phases (2 and 4%) and G2/M (93%), respectively. (Figure 3C,D and Figure 4). The percentage of inhibition of proliferation of HepG2 treated with 30 µg/mL 6-MP and 5 µg/mL 6-MP loaded positive liposomes was found to be 65% and 80%, respectively, whereas the percentage of inhibition of HepG2 treated positive liposome was 7%. HepG2 suppression by 6-MP or 6-MP liposomes occurs because 6-MP inhibited de novo ribonucleotide synthase and purine ribonucleotide interconversion and cooperated with DNA structure after 6-MP converted into 6-thioguanine. When 6-thioguanine is incorporated into DNA, DNA-protein crosslinks and single-strand breaks result. Thiopurine also causes the level and activity of the enzyme DNA (cytosine-5)-methyltransferase (DNMT1) 1 to decrease, which lowers DNA methylation [40].

In the present investigation, HepG2 treated with 30 µg/mL 6-MP had greater percentages of cells in the S phase (21.8%) and G2/M (32.6%) than untreated HepG2 did: (17.3%) and (27.5%), respectively. In cells, 6-MP is converted to 6-thioguanine (6-TG), which cooperated with DNA. The DNA Mismatched Repair (MMR) system is triggered, and S-adenosylmethionine methylates the 6-TG nucleotide in MMR. Thyamine (T) and 6-TG deoxynucleotide base pairs (6-TG:T) activate MMR and cause cell death. Cells treated with 6-MP are arrested in the S-phase because DNA damage and proliferation take place during the S-phase [41]. According to a different study, 6-TG-treated cells undergo post-replication DNA damage in the G2 phase. Cells that have received 6-TG for a long period of time continue to be arrested in the G2 phase [42].

The percentage of G2/M (32.6%) and sub-G1 (32.3%) cells in cells treated with 30 µg/mL 6-MP and 5 µg/mL 6-MP loaded positive liposomes, respectively, in the current investigation. ATR activations caused by DNA damage after MMR activation phosphorylate and activate check point kinase (CHK1) in the G2 phase. Cells treated with 6-MP LP liposomes exhibit cell cycle arrest in subG1 as a result of extensive DNA damage [43]. Additionally, ATM triggers CHK2, which arrests the cell cycle in the M phase when the double strand DNA is broken. The ATR-ATM response is enhanced by the redirection of the transducer checkpoint kinases (CHK1 and CHK2). In acute promyelocytic leukemia, CHK2 phosphorylates ATM to help phosphorylate PML protein or binds to p53 to cause apoptosis [44].

Many researchers have synthesized different 6-MP loaded nanoparticles and investigated their activity. Dorniani et al. prepared iron oxide nanoparticles coated with chitosan and 6-mercaptopurine. They reported that these nanoparticles were not considered to be toxic to a normal mouse fibroblast cell after performing the MTT assay [20]. Other workers developed 6-MP gold nanoparticles and mentioned an improvement in the antiproliferative activity against K-562 leukemia cells [38]. Lu et al. designed hyaluronic acid-based gluteal-skin-responsive 6-MP polymer prodrug for highly effective targeted treatment of acute myeloid leukemia [45]. Our work is different in the carrier system and in the studied cell lines.

## 4. Conclusions

In the current work, 6-MP encapsulated liposomes were successfully developed and showed nanosized vesicles with a good size distribution and drug entrapment efficiency. Incorporation of stearyl amine in the formulation results in development of positively charged liposomal formulation which is expected to physically conjugate with the negative charges of cancer cells’ cell membranes. Despite having a lower entrapment efficiency than F2, 6-MP loaded positive liposomes (F1) had an IC_50_ that was nearly two times lower against HepG2, HCT116, and MCF-7. In HepG2 treated with F1, there were no signs of apoptosis. Treatment of HepG2 with F1 resulted in the highest percentage of inhibition. Inhibit of HepG2 cell growth was found to be in a concentration-dependent manner. Because of DNA damage that started in the G2 phase and continued to the sub-G1 phase, HepG2 cells treated with 6-MP liposomes (F1) were stopped in the sub-G1 and G2/M phases, which inhibited cells from proliferating. The nanosized liposome loaded 6-MP may have a better absorption than the pure drug, improving bioactivity at lower drug concentrations, and potentially lowering the likelihood of adverse 6-MP effects. In conclusion, 6-MP encapsulated within the liposomes carrying the stearyl amine may be beneficial against a variety of cancers but the stability of this formulation in cell medium or blood, pharmacokinetics, and in vivo studies are required in order to consider 6-MP loaded liposomal formulation as a novel commercial drug.

## Figures and Tables

**Figure 1 nanomaterials-12-04029-f001:**
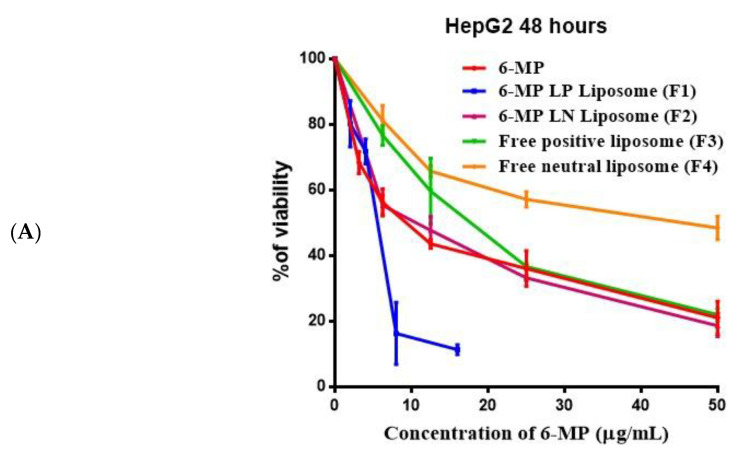
Percentage of cell viability of HepG2 (**A**), HCTT_116_ (**B**), and MCF-7 (**C**) after treatment with different concentrations of free 6-MP and the prepared liposomal formulations.

**Figure 2 nanomaterials-12-04029-f002:**
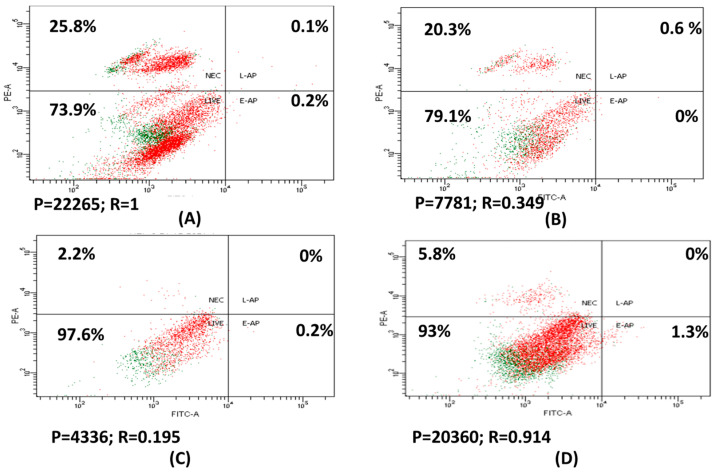
HepG2 staining with Annexin V/7-PI; control (**A**); treated with 30 µg/mL 6-MP (**B**); 5 µg/mL 6-MP loaded with positive charge liposome [F1] (**C**); and free positive charge liposomes [F3] (**D**).

**Figure 3 nanomaterials-12-04029-f003:**
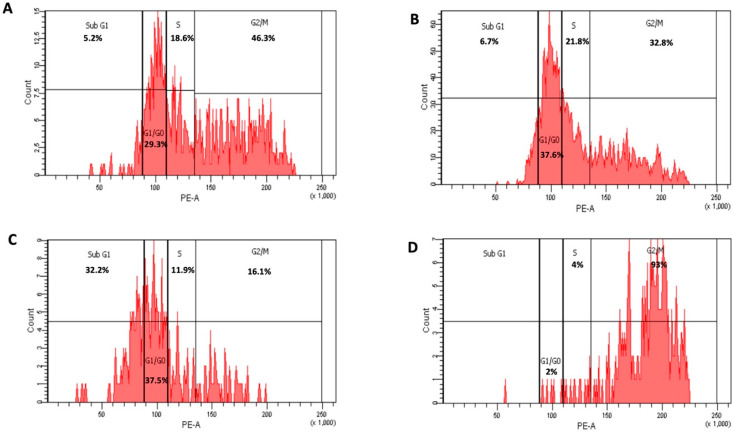
Cycle arrest of untreated HepG2, control (**A**); treated with 30 µg/mL 6-MP) (**B**); 5 µg/mL of 6-MP loaded positive charge liposomes [F1] (**C**); and drug free positive charge liposomes [F3] (**D**).

**Figure 4 nanomaterials-12-04029-f004:**
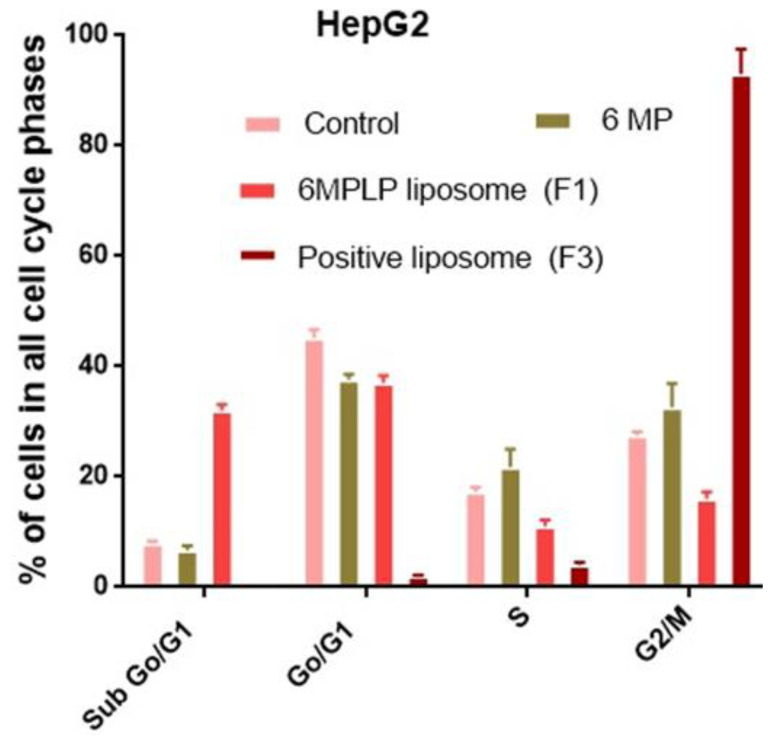
Percentage of cells in all cell cycle phases in HepG2 treated with free 6-MP and loaded with positive charged liposome.

**Table 1 nanomaterials-12-04029-t001:** Formulation ingredients and characterization of the prepared liposomal vesicles.

Run	Ingredients		EE (%)	Drug Loading (%)	Size (nm)	PDI	ZP (mV)
	Drug	Stearyl Amine					
F1	√	√	42.11 ± 2.07	3.56 ± 0.07	574.67 ± 37.29	0.543 ± 0.031	+12.4 ± 1.31
F2	√	-	94.44 ± 0.56	9.42 ± 0.21	660.47 ± 44.32	0.667 ± 0.046	−0.48 ± 0.13
F3	-	√	-	-	429.47 ± 24.79	0.465 ± 0.052	+9.61 ± 0.69
F4	-	-	-	-	538.80 ± 49.73	0.711 ± 0.049	−0.74 ± 0.24

**Abbreviations**: F1, positively charged 6-MP-loaded liposomes; F2, neutral 6-MP-loaded liposomes; F3, non-medicated positively charged liposomes, and F4, non-medicated uncharged liposomes. EE, entrapment efficiency; PDI, polydispersity index; and ZP, zeta potential.

**Table 2 nanomaterials-12-04029-t002:** IC_50_ of the prepared liposomal formulations on HepG2, HCT116, and MCF-7 human cell lines after 48 h.

Cells	IC_50_ (Range/Value)	Pure Drug	F1	F2	F3	F4
HepG2	IC_50_ range Value of IC_50_ (µg/mL)	8.0–11.6 9.6	2.8–7.6 4.6	8.9–12.1 10.35	14.18–19.68 16.7	27.8–41.4 33.9
HCT116	IC_50_ range Value of IC_50_ (µg/mL)	14.0–19.9 16.7	4.0–5.6 4.70	12.4–17.2 14.6	13.7–18.9 16.12	32.1–42.9 37.1
MCF7	IC_50_ range Value of IC_50_ (µg/mL)	10.7–15.2 12.8	3.8–6.4 4.9	10.3–13.2 11.6	19.0–24.4 21.5	36.0–48.9 41.9

## Data Availability

Not applicable.

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
