# Peer review of "Preparation of 6-Mercaptopurine Loaded Liposomal Formulation for Enhanced Cytotoxic Response in Cancer Cells"

_nanomaterials, 2022, doi:10.3390/nano12224029_

Round 1
Reviewer 1 Report
This is an interesting study about 6-mercaptopurine-loaded liposomal formulation for enhanced cytotoxic response in cancer cells. It could be publishable after the following points are well addressed.
1. The drug loading content should be determined.
2. The quality of the figures should be improved to a higher level.
3. Long stability of the formulation in cell medium or blood is encouraged to be added.
4. Line 68-72, several studies (Acta Biomaterialia 136 (2021) 558–569; Biomacromolecules 2016, 17, 6, 2010–2018; Colloids and Surfaces B: Biointerfaces 188 (2020) 1107) are encouraged to be included to support such a claim.
5. Formatting issues. Figure 1, only HepG2 results are shown. Please check all.
Author Response
This is an interesting study about 6-mercaptopurine-loaded liposomal formulation for enhanced cytotoxic response in cancer cells. It could be publishable after the following points are well addressed.
- The drug loading content should be determined.
Reply
The drug loading has been calculated and added to the revised manuscript.
- The quality of the figures should be improved to a higher level.
Reply
More clear figures have been added to the revised manuscript. Moreover, all the manuscript figures have been submitted as a separate file in high quality resolution.
- Long stability of the formulation in cell medium or blood is encouraged to be added.
Reply
We highly appreciate the reviewer comments about the long term stability of the formulation in cell medium or blood. This part will be studied and submitted as a separate work along with the pharmacokinetic and in vivo studies. This explanation has been added to the revised manuscript (conclusion section).
- Line 68-72, several studies (Acta Biomaterialia 136 (2021) 558–569; Biomacromolecules 2016, 17, 6, 2010–2018; Colloids and Surfaces B: Biointerfaces 188 (2020) 1107) are encouraged to be included to support such a claim.
Reply
The suggested references have been added to the revised manuscript.
- Formatting issues. Figure 1, only HepG2 results are shown. Please check all.
Reply
Figure 1 has been modified to provide all results for the studied cell lines.
Reviewer 2 Report
Generally, the authors present an interesting study. The paper is well prepared. However, some corrections are needed. Please find my specific comments below:
1. Line 25, ‘6MP’ should be ‘6-MP’.
2. Line 62-63, please cite the literature which proved the ‘The half-life of 6-MP is ranged between 0.5 and 1.5 hours’. Does it mean plasma half-life?
3. Line 75, please double check the sentence ‘Lipophilic and lipophilic drugs are encapsulated in liposomes’.
4. Line 89-90, ‘IC50’ should be ‘IC50’.
5. Line106-107, I would suggest the authors add some sentences to describe the ‘Thin-film hydration technique’. What was the advantage or disadvantage of this technique?
6. Line 119-120, I would suggest the authors remove the ethanol at reduced pressure. It is impossible to remove all ethanol at 50 ºC under normal pressure.
7. Line 138-139, the description of the experiment was not clear. Is any literature support this method? Please cite the related literatures.
8. Line 150,154,157, I do not think ‘100L’ was correct.
9. Line 153, ‘50, 25, 12, 5, 6.25, and 3.125 g/ml’ should be ‘50, 25, 12.5, 6.25, and 3.125 g/ml’.
10. Line-157-158, the description of the experiment was not clear.
11. ‘300L’? Please double check it.
12. Line 248, ‘stearyl mine [CH3 (CH2)17NH2]’ should be ‘stearyl amine [CH3 (CH2)17NH2]’. Line 454 (maybe other place), ‘stearyl mine’ should be ‘stearyl amine’.
13. Line 287, Figure 1, figure B and C were missing.
14. Line 260-261, the authors mentioned ‘The IC50 values for free positive and neutral liposomes (F3 and F4) in HepG2, HCT116, and MCF-7 were (16.7, 33.9), (16.2, 37.1), and (21.5, 41.9), respectively’. However, from figure A in Fig 1 (the only figure I can see in Fig 1), the IC50 value for F3 in HepG2 should be higher than 50, not 16.7 as the paper calculated. Please double check all IC50 values to make sure all they are correct.
15. Line 298, please double check the name of the compound.
16. Line 316, ‘HCT116’ should be ‘HCT116’.
17. Some people have encapsulated 6-MP using other methods. I would suggest the authors compare their research work with other people’s work and do some in-depth discussion.

Author Response
Generally, the authors present an interesting study. The paper is well prepared. However, some corrections are needed. Please find my specific comments below:
- Line 25, ‘6MP’ should be ‘6-MP’.
Reply
Correction done.
- Line 62-63, please cite the literature which proved the ‘The half-life of 6-MP is ranged between 0.5 and 1.5 hours’. Does it mean plasma half-life?
Reply
The elimination half-life of 6-MP is ranged between 1 -2 hours.
Reference has been cited.
- Line 75, please double check the sentence ‘Lipophilic and lipophilic drugs are encapsulated in liposomes’.
Reply
The sentence has been modified. More explanation has been included. Reference has been added.
- Line 89-90, ‘IC50’ should be ‘IC50’.
Reply
The symbol has been corrected allover the manuscript.
- Line106-107, I would suggest the authors add some sentences to describe the ‘Thin-film hydration technique’. What was the advantage or disadvantage of this technique?
Reply
We highly appreciate the reviewer comments. The required information has been added to the revised manuscript.
- Line 119-120, I would suggest the authors remove the ethanol at reduced pressure. It is impossible to remove all ethanol at 50 ºC under normal pressure.
Reply
Sentence has been modified.
- Line 138-139, the description of the experiment was not clear. Is any literature support this method? Please cite the related literatures.
Reply
The sentence has been modified to make it more clear to readers. Reference has been added.
- Line 150,154,157, I do not think ‘100L’ was correct.
Reply
We apologize for this typo mistake. The correct information has been included in the revised manuscript.
- Line 153, ‘50, 25, 12, 5, 6.25, and 3.125 g/ml’ should be ‘50, 25, 12.5, 6.25, and 3.125 g/ml’.
Reply
Correction has been done.
- Line-157-158, the description of the experiment was not clear.
Reply
The sentence has been modified to make it more clear for readers.
- ‘300L’? Please double check it.
Reply
Sentence has been corrected.
- Line 248, ‘stearyl mine [CH3(CH2)17NH2]’ should be ‘stearyl amine [CH3 (CH2)17NH2]’. Line 454 (maybe other place), ‘stearyl mine’ should be ‘stearyl amine’.
Reply
Sentence has been corrected.
- Line 287, Figure 1, figure B and C were missing.
Reply
Figure 1 has been modified and added to the manuscript.
- Line 260-261, the authors mentioned ‘The IC50values for free positive and neutral liposomes (F3 and F4) in HepG2, HCT116, and MCF-7 were (16.7, 33.9), (16.2, 37.1), and (21.5, 41.9), respectively’. However, from figure A in Fig 1 (the only figure I can see in Fig 1), the IC50 value for F3 in HepG2 should be higher than 50, not 16.7 as the paper calculated. Please double check all IC50 values to make sure all they are correct.
Reply
The sentence has been rephrased. Figure 1 has been uploaded correctly. Values of IC50 has been checked and stated in the text based on the calculated values in table 2.
- Line 298, please double check the name of the compound.
Reply
The name of the compound has been corrected.
- Line 316, ‘HCT116’ should be ‘HCT116’.
Reply
Sentence has been corrected.
- Some people have encapsulated 6-MP using other methods. I would suggest the authors compare their research work with other people’s work and do some in-depth discussion.
Reply
Other systems that have encapsulated 6-MP have been mentioned and we have compared their results to our work. More discussion has been included in the revised manuscript.
Reviewer 3 Report
This work entitled "Preparation of 6-mercaptopurine loaded liposomal formulation for enhanced cytotoxic response in cancer cells". Thus, this reviewer recommends the publication of this work after addressing the following comments.
Zeta potential measurements – provide the applied scattering angle, the analysis time, the number of runs and scan for each run, the dilution ratio, and the refractive index of each sample.
The authors are encouraged to provide all the materials used to perform the experiments and analysis, along with their respective suppliers. Moreover, Please report the molecular weight, this is relevant information to ensure appropriate reproducibility.
The conclusion must be revised and answer the hypothesis of the work, and it should not be similar to the abstract.
Author Response
This work entitled "Preparation of 6-mercaptopurine loaded liposomal formulation for enhanced cytotoxic response in cancer cells". Thus, this reviewer recommends the publication of this work after addressing the following comments.
- Zeta potential measurements – provide the applied scattering angle, the analysis time, the number of runs and scan for each run, the dilution ratio, and the refractive index of each sample.
Reply
All the required information for zeta potential measurement have been added to the revised manuscript.
- The authors are encouraged to provide all the materials used to perform the experiments and analysis, along with their respective suppliers. Moreover, Please report the molecular weight, this is relevant information to ensure appropriate reproducibility.
Reply
The materials section has been revised and all required information has been added.
- The conclusion must be revised and answer the hypothesis of the work, and it should not be similar to the abstract.
Reply
The conclusion section has been revised.
Round 2
Reviewer 1 Report
The quality of all the figures still needs to be improved. For example, you can't see the text and numbers in Figures 2 and 3 clearly.
Author Response
Reviewer 1
The quality of all the figures still needs to be improved. For example, you can't see the text and numbers in Figures 2 and 3 clearly.
Reply
More clear figures have been uploaded to the modified manuscript.
Reviewer 2 Report
The authors have tried hard to adjust the text to be clearer. Most questions raised by the reviewers are well answered. I believe the manuscript has been significantly improved and now warrants publication after some corrections.1. Line 186,I see that ‘μl’ and ‘μL’ were used in one sentence. I would suggest the authors choose either one, also take care of ‘ml’ and ‘mL’ in the whole manuscript.
2. Line 280-281, why in Fig A of Fig 1, yellow line represents F3 and green line represents F4; however, in Fig B and C of Fig 1, yellow line represents F4 and green line represents F3? I thought the authors have made a mistake in Fig A. In my former review, I indicated the IC50 value of F3 (16.7, yellow line) in Fig A maybe wrong because I only can see Fig A, but Fig B and C were missing. In this version of manuscript, 16.7 should be the IC50 value of green line and it looks ok. However, visually judged from the Fig A, B and C in Fig 1, it seems all the IC50 values for the yellow line were not correct. Because I do not have the experimental data, I would suggest the authors to carefully double check all the IC50 values for the yellow line to make sure the results are correct.

Author Response
Reviewer 2
The authors have tried hard to adjust the text to be clearer. Most questions raised by the reviewers are well answered. I believe the manuscript has been significantly improved and now warrants publication after some corrections.
- Line 186,I see that ‘μl’ and ‘μL’ were used in one sentence. I would suggest the authors choose either one, also take care of ‘ml’ and ‘mL’ in the whole manuscript.
Reply
The units have been corrected in the whole manuscript.
- Line 280-281, why in Fig A of Fig 1, yellow line represents F3 and green line represents F4; however, in Fig B and C of Fig 1, yellow line represents F4 and green line represents F3? I thought the authors have made a mistake in Fig A. In my former review, I indicated the IC50 value of F3 (16.7, yellow line) in Fig A maybe wrong because I only can see Fig A, but Fig B and C were missing. In this version of manuscript, 16.7 should be the IC50 value of green line and it looks ok. However, visually judged from the Fig A, B and C in Fig 1, it seems all the IC50 values for the yellow line were not correct. Because I do not have the experimental data, I would suggest the authors to carefully double check all the IC50 values for the yellow line to make sure the results are correct.
Reply
We apologize for this mistake.. Figure 1 has been updated and uploaded. Also, values for IC50 have been reviewed.